# LUT Based Neural Networks as Neuro-Symbolic Systems

**Lizy K. John**                                                        LJOHN@ECE.UTEXAS.EDU
*The University of Texas at Austin, USA*

**Priscila M. V. Lima**                                        PRISCILAMVL@COS.UFRJ.BR
*Federal Univeristy of Rio de Janeiro, BR*

**Alan T. L. Bacellar**                                        ALANBACELLAR@UTEXAS.EDU
*The University of Texas at Austin, USA*

**Shashank Nag**                                                  SHASHANKNAG@UTEXAS.EDU
*The University of Texas at Austin, USA*

**Eugene B. John**                                                EUGENE.JOHN@UTSA.EDU
*The University of Texas at San Antonio, USA*

**Felipe M. G. Franca**                                                  FELIPE@IEEE.ORG
*IT-Universidade do Porto (now at Google), LLC*

**Editors:** Leilani H. Gilpin, Eleonora Giunchiglia, Pascal Hitzler, and Emile van Krieken

Although not a recent topic of research, neurosymbolic systems (NSAI) (Hitzler and Sarker, 2022; Besold et. al, 2022; Lima, 2001; van Harmelen, 2022) may have been one of the latest frontiers in artificial intelligence to catch the attention of a broader scientific community. One possibility for such interest could be that NSAI is a fundamental step towards artificial general intelligence (AGI). Many would argue that a tightly coupled integration of neural and symbolic paradigms would not be necessary, as the state-of-the-art of both sides could interact through a common interface. Others, however, may see the benefits of having a tight integration under the same computational substrate. Such combination of skills comes, however, with a computational price, both in terms of memory and time costs. When it comes to the point of deploying these hybrid models in silicon, these computational costs may constitute a serious drawback, especially with respect to online learning. This work proposes the adoption of a family of weightless neural networks (WNNs)(Aleksander et al., 2009) to bring neurosymbolic systems to the level of integrated circuits.

WNNs are a distinct class of neural models which derive inspiration from the decoding process issued by the dendritic trees of biological neurons. Instead of weights and dot products to determine neural activity, they utilize look up tables (LUTs). An n-input LUT can hold any one of $2^{2^n}$ possible logic functions, resulting in significant learning capacity (Carneiro et al., 2019) compared to models based on multiply add operations. WNNs are inherently low-energy and low latency since primarily only table lookup is involved, and can easily be prototyped/fabricated in hardware. Our initial FPGA prototypes of LUT node based WNNs (Susskind et al., 2022) with Counting Bloom filters, arithmetic-free hashing, and with bleaching consume 85-99% fewer cycles and 80-95% less energy compared to deep neural networks of the same accuracy. We have further improved WNNs by ensembles and pruning of LUT nodes, and ULEEN (Susskind et. al., 2023) can excel over BNNs (Umuroglu et al., 2017).

Our recent research has created Differentiable Weightless Neural Networks (DWNs) (Bacellar et al., 2024) using principles of Extended Finite Differences (EFD). We also employ Learnable Mapping, Learnable Reduction, and Spectral Regularization to improve the accuracy and reduce the model size and efficiency. On several workloads including Keyword Spotting and Anomaly Detection from MLPerfTiny, DWN provides 10X throughput and better accuracy versus AMD/Xilinx FINN implementations. On 11 tabular datasets, DWN yielded more accuracy and higher throughput, but more notably yielded very tiny classifiers, smaller than the classifiers yielded by DiffLogicNet (Petersen et al., 2022) and Tiny Classifiers (Iordanou et. al., 2024). In software implementations, DWN compares favorably to implementations from AutoGluon XGBoost/CatBoost/LightGBM/TabNN/NNFastAITab (Erickson et al., 2020) and Google TabNet. The most surprising observation was that on a few datasets the DWN training/input mapping methodology yielded near-zero hardware implementations, suggesting that DWNs have some unique ability in extracting symbols. DWNs can be considered as a symbol extractor, or it can be an ultra-fast ultra-thin neurosymbolic inference engine. The learnable input mapping can be considered similar to rule-based learning and the Look Up Table contents can be considered as the neural component. In DWN, the integration of explicit knowledge with that implicitly acquired, in a similar fashion to other weightless models (Ludermir et al., 2008), is the subject of ongoing research.

## Acknowledgments

This research was supported in part by Semiconductor Research Corporation (SRC) Task 3148.001, National Science Foundation (NSF) Grants #2326894 & #2425655 and NVIDIA Applied Research Accelerator Program Grant. Any opinions, findings, conclusions, or recommendations are those of the authors and not of the funding agencies.

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
