# OpenReview forum: "LUT Based Neural Networks as Neuro-Symbolic Systems"
_nesyconf.org/NeSy/2025/Conference_Phase_2 — NeSy 2025 - Phase 2 Poster_

### Official Review · Reviewer_MKbf · 2025-06-26
**Valuable contribution to efficient neural modeling with limited connection to symbolic reasoning**

**Rating:** 6
**Confidence:** 3

**Review:**

This extended abstract presents Differentiable Weightless Neural Networks (DWNs), an efficient neural architecture built on lookup tables. The approach incorporates Learnable Mapping, Learnable Reduction, and Spectral Regularization to enhance accuracy and reduce model size. DWNs were evaluated on 11 tabular datasets, demonstrating strong performance and producing very compact classifiers. The work is well-motivated and provides useful context, but given the limited space, I would suggest reducing some of the broader background in favor of more detail on the approach itself, particularly the details of the DWN architecture. Additionally, more clarity on the domains of the tabular datasets and whether keyword spotting and anomaly detection were the only tasks considered would strengthen the paper.

While the work is interesting and the contributions are technically solid, in my view, this paper appears to be focused on a purely neural architecture based on differentiable lookup tables. I do not observe any explicit symbolic reasoning or neuro-symbolic integration as typically understood in the literature. Although the authors suggest that DWNs could serve as symbol extractors or inference engines—drawing an analogy between learnable mappings and rule-based learning—this remains a speculative interpretation. There is no formal use of symbolic representations or logic-based reasoning mechanisms. Therefore, I consider this work to be outside the intended scope of the NeSy conference.

**Anonymity:**

Remain anonymous

---

### Official Review · Reviewer_LRwF · 2025-07-04

**Rating:** 6
**Confidence:** 4

**Review:**

This paper introduces Differentiable Weightless Neural Networks (DWNs) as a neurosymbolic computing architecture, building on the lineage of weightless neural networks (WNNs). DWNs rely on lookup tables (LUTs) and Extended Finite Differences (EFD) to perform differentiable learning without the need for multiply-accumulate operations. The authors evaluate DWNs on a variety of tasks such as tabular datasets, keyword spotting, and anomaly detection, showcasing competitive performance in both software and hardware implementations. The work highlights DWNs' potential for low-latency, low-energy inference and their symbolic representation capabilities.

Pros
+ The neurosymbolic field is gaining traction as researchers seek efficient models that combine reasoning and perception. The use of DWNs in this context is well-motivated. The authors provide compelling claims of DWNs outperforming conventional deep models in energy and latency, which is particularly relevant for edge inference.

+ Good introduction to DWNs: The paper provides a clear overview of LUT-based networks and the evolution to differentiable variants.

Cons
- The phrase “Our recent research has created...” directly hints at authorship and violates the anonymity guidelines of the double-blind review process. This should be rephrased or anonymized.

**Anonymity:**

Remain anonymous

---

### Official Review · Reviewer_tU7R · 2025-07-07
**Differentiable Weightless Neural Networks**

**Rating:** 6
**Confidence:** 3

**Review:**

This extended abstract presents the Differentiable Weightless Neural Network (DWN), which is a hardware-efficient model.
The authors draw a connection between their model's components and the principles of neuro-symbolic AI, framing the DWN as a system with a rule-based component (the learnable input mapping) and a neural component (the lookup tables (LUTs)).
The work fits in the scope of the conference, highlighting fresh perspectives, particularly on the importance of reducing the computational costs often brought by neuro-symbolic methods.

**Anonymity:**

Remain anonymous